# A Dietary Intervention of Bioactive Enriched Foods Aimed at Adults at Risk of Metabolic Syndrome: Protocol and Results from PATHWAY-27 Pilot Study

**DOI:** 10.3390/nu11081814

**Published:** 2019-08-06

**Authors:** Achim Bub, Corinne Malpuech-Brugère, Caroline Orfila, Julien Amat, Alice Arianna, Adeline Blot, Mattia Di Nunzio, Melvin Holmes, Zsófia Kertész, Lisa Marshall, Imola Nemeth, Luigi Ricciardiello, Stephanie Seifert, Samantha Sutulic, Marynka Ulaszewska, Alessandra Bordoni

**Affiliations:** 1Department of Physiology and Biochemistry of Nutrition, Max Rubner-Institut, 76131 Karlsruhe, Germany; 2Université Clermont Auvergne, INRA, UNH, Unité de Nutrition Humaine, CRNH Auvergne, F-63000 Clermont-Ferrand, France; 3School of Food Science and Nutrition, University of Leeds, Leeds LS2 9JT, UK; 4CHU Clermont Ferrand, CRNH Auvergne, F-63000 Clermont-Ferrand, France; 5Department of Agri-Food Sciences and Technologies (DISTAL)—University of Bologna (IT) Piazza Goidanich, 60, 47521 Cesena (FC), Italy; 6Campden BRI (Hungary Site), Haller Str 2, 1096 Budapest, Hungary; 7AdWare Research Ltd., Völgy u. 41, 8230 Balatonfüred, Hungary; 8Department of Medical and Surgical Sciences, University of Bologna, Via Massarenti 9, 40138 Bologna, Italy; 9Dipartimento Qualità Alimentare e Nutrizione, Centro Ricerca ed Innovazione-Fondazione Edmund Mach, 38010 San Michele all’Adige, Italy

**Keywords:** nutritional intervention trials, functional food, metabolic syndrome, docosahexaenoic acid, oat beta-glucan, anthocyanins, bioactive, short chain fatty acid, egg, bakery, dairy

## Abstract

Around a quarter of the global adult population have metabolic syndrome (MetS) and therefore increased risk of cardiovascular mortality and diabetes. Docosahexaenoic acid, oat beta-glucan and grape anthocyanins have been shown to be effective in reducing MetS risk factors when administered as isolated compounds, but their effect when administered as bioactive-enriched foods has not been evaluated. Objective: The overall aim of the PATHWAY-27 project was to evaluate the effectiveness of bioactive-enriched food consumption on improving risk factors of MetS. A pilot study was conducted to assess which of five bioactive combinations provided within three different food matrices (bakery, dairy or egg) were the most effective in adult volunteers. The trial also evaluated the feasibility of production, consumer acceptability and gastrointestinal tolerance of the bioactive-enriched food. Method: The study included three monocentric, parallel-arm, double-blind, randomised, dietary intervention trials without a placebo. Each recruiting centre tested the five bioactive combinations within a single food matrix. Results: The study was completed by 167 participants (74 male, 93 female). The results indicated that specific bioactive/matrix combinations have effects on serum triglyceride or HDL-cholesterol level without adverse effects. Conclusion: The study evidenced that bioactive-enriched food offers a promising food-based strategy for MetS prevention, and highlighted the importance of conducting pilot studies.

## 1. Introduction

Metabolic syndrome (MetS) is the name given to a cluster of conditions that occur together more often than can be explained by chance. In 2009, a joint statement was published by world-leading authorities on cardiovascular disease and diabetes that harmonised the criteria by which MetS is diagnosed [1]. It is estimated that 20%–25% of the global adult population have MetS [2]. People with MetS are three times as likely to suffer a heart attack or stroke and are twice as likely to die from these causes, compared with people who do not have MetS [3]. In addition, people with MetS have a fivefold greater risk of developing type-2 diabetes [4]. Therefore, lifestyle interventions including dietary modification to reduce the prevalence of MetS are urgently needed. Some food bioactives have been shown to impact on one or more MetS risk factors through various mechanisms. However, the natural abundance of most bioactives in food and therefore their intake within habitual diets are generally lower than the optimally effective dose [5]. Many studies have used pure bioactives in supplement form to show effectiveness [6,7]. However, supplementation may not be a sustainable or cost-effective strategy to improve public health. Delivering a higher concentration of bioactives than normally achieved through the diet may be achieved through formulation of bioactive enriched foods (BEFs) [8]. If these BEF could be commercially available, then they could be regularly consumed as part of habitual diet.

In addition, food enrichment could allow food companies to exploit the already approved health claims for bioactives [9]. The use of health claims on food in Europe is regulated by the European Food Safety Authority (EFSA) and application for the use of a health claim requires substantial evidence to demonstrate a cause and effect relationship between consumption of the functional food and a physiologically beneficial health outcome. To date, only 11% of health claim applications have been successful, including a favorable opinions for docosahexaenoic acid (DHA) [10] and beta glucan [11]. Gathering evidence to substantiate a health claim is time-consuming and expensive, and frequently cited as a barrier for small and medium sized enterprises (SMEs) wishing to develop a functional food with an approved health claim [9,12]. Trials with foods pose particular technical, economic and operational challenges compared to supplements [13]. Challenges include the manufacture of large quantities of perishable test foods in specialized food grade facilities, often at high cost. For these reasons, feasibility or pilot studies can be useful as preliminary studies to assist the design of larger trials. Eldridge et al. (2015) defined a conceptual framework for feasibility and pilot trials [14]. As explained by Marsaux et al. (2017), a feasibility study assesses if research or activity can be done, should be done, and if so, how [15]. A pilot study answers similar questions but as part of a future study. A pilot study is useful to test or select parameters for a larger trial. In this framework, pilot studies are thus a subset of feasibility studies [15].

PATHWAY-27 is a project involving a multidisciplinary team of scientists from academia and industry (http://pathway27.eu/) with the aim of developing dietary interventions with BEFs, and specifically to test the effectiveness of BEFs containing DHA, anthocyanins (AC), or oat beta-glucan (OBG) on improving risk factors of MetS. These bioactives were selected based on their putative beneficial effects on markers of MetS [16]. They were used as ingredients of BEF in three different widely consumed food matrices (dairy, egg and bakery products). The overall aim of the PATHWAY-27 project was to evaluate the effectiveness of bioactive-enriched food consumption on improving risk factors of MetS. A pilot study was conducted to assess which of five bioactive combinations provided within three different food matrices (bakery, dairy or egg) were the most effective in adult volunteers. The trial also evaluated the feasibility of production and distribution of food, consumer acceptability, blinding, recruitment flow, measurements of compliance and gastrointestinal tolerance of the bioactive-enriched food. The study considered the potential synergisms between DHA and the other two bioactives, as well as the influence of the food matrix on bioactive effectiveness on MetS markers. This paper describes the design, implementation and results of the Pathway-27 multi-centric pilot study.

## 2. Materials and Methods

### 2.1. Chemicals

All chemicals and solvents were from Sigma Aldrich (Gillinham, UK) unless specified.

### 2.2. Participants

Healthy men and women aged 18 to 80 years were eligible to take part in the study if they presented with 2, 3 or 4 of the criteria for MetS diagnosis [17], with at least one of them being elevated fasting triglycerides (TG) or low high density lipoprotein cholesterol (HDL-C). Major exclusion criteria are reported in [18]. As this was a pilot study, no formal power calculation was performed. The study aimed to recruit at least 10 participants per enrichment, per matrix.

### 2.3. Pilot Trial Design

The PATHWAY-27 pilot study was conducted as three monocentric parallel-arm, double-blind, randomised trials without placebo in three recruiting centres (RC): Centre de Recherche en Nutrition Humaine Auvergne, Clermont-Ferrand, France (CRNH); Max Rubner-Institut, Karlsruhe, Germany (MRI); and School of Food Science and Nutrition, University of Leeds, Leeds, UK (ULE). The study was performed in accordance with the ethical guidelines of the Declaration of Helsinki. All participants provided written informed consent. Ethical approval for the study protocol was obtained from national authorities: MRI ethical committee approval reference number: F-2014-062 (State Medical Chamber Baden-Wuerttemberg); CRNH Regional Committee ethics reference number: 2014-A01290-47; ULE MEEC-Ethics reference number: MEEC 13-027, amended reference number MEEC 14-017. Design of the intervention and summary of participant activities are illustrated in Figure 1. The protocol has been made open access [18]. The egg arm of the trial was registered in www.clinicaltrials.gov reference NCT03956433.

### 2.4. Test Foods

DHA, AC and OBG enrichment was obtained by adding OVO-DHA^®^ (Applications Sante des Lipides Sarl, Hauterive, France), Eminol^®^ (ABRO BIOTEC SL, Valbuena de Duero, Spain) and SweOat^®^ bran BG28 XF (Swedish Oat Fiber, Bua, Sweden), respectively. Pancake and milkshake were manufactured by production plants coordinated by ADEXGO Kft (Lapostelki utca, Hungary), and biscuits by Desarrollos Panaderos Levantinos SLL (Valencia, Spain).

Each RC tested five enrichments within one food matrix: bakery BEF were tested at CRNH in the form of ready to eat enriched biscuits; dairy BEF at MRI in the form of a powdered milkshake which was reconstituted with water by the participant in re-usable shaker provided, following detailed preparation instructions; and egg BEF at ULE in the form of frozen pancakes which were reheated by the participant. Participants were randomly allocated to one of five parallel treatment arms and required to eat one portion of the experimental food each day for four weeks. Each arm corresponded to enrichment types as shown in Table 1 and the energy and macronutrient composition of the food matrices is found in Table 2. The dose of DHA was set close to the adequate intake (AI) dose for adults set by EFSA [19]. The dose for anthocyanin was chosen based on published research reporting beneficial effects of anthocyanins consumption on LDL-C and HDL-C metabolic markers [7]. Finally, the dose of OBG was chosen according the EFSA opinion on the health claim for OBG and the lowering of blood cholesterol [20]. Participants were asked to consume the randomly allocated BEF in replacement of similar foods that they would usually have, not in addition to their normal diet. Volunteers were asked to avoid modifying their lifestyle and dietary habits, except for the restriction to one portion per day of foods having a high content of the bioactives being investigated. A list of restricted foods was provided to volunteers [18]. Blinding was achieved though packaging of foods in opaque foil packaging (Appendix A) and labelling of foods with a specific code for each participant. Chemical analysis of bioactive content and microbiological analysis was performed on each production batch.

### 2.5. Randomization

In order to achieve equal group sizes amongst the treatment arms across all RC, a block randomization method was applied. Each randomization list consisted of a three-digit randomization number from 101 to 160 for male participants and from 201 to 260 for female participants, with the name and code of the assigned enrichment material. Each recruitment centre was also allocated a unique one-digit identification code. So, volunteers were identified by a four-digit randomisation code comprised of the one-digit centre ID plus three-digit randomisation number. The randomisation lists were sent to the food manufacturer of each food product, who packed and labelled the products according to the list and distributed the food products to the RC. Room temperature distribution was applied to bakery and dairy BEF, while frozen-chain was necessary for the egg BEF.

### 2.6. Blinding

The intervention was double-blinded. Products were packaged in opaque foil packaging (Appendix A) and labelled with the randomization code, expiry date and eating instructions. Throughout the trial blinding was maintained since no serious adverse effects occurred.

### 2.7. Blood Pressure and Anthropometric Measurements

At each visit to the RC, blood pressure was measured in participants after being at rest for five minutes in a seated position followed by two additional measurements separated by two minutes each using a calibrated sphygmomanometer and appropriate sized cuffs after measurement of arm circumference. Body weight and height were measured to the nearest 0.1 cm and BMI was calculated according to WHO guidelines [21]. All measurements were performed by trained study researchers in strict adherence to standard operating procedures (SOPs) harmonised for all three RC as described in [18].

### 2.8. Blood Collection, Processing and Analysis

Blood was sampled by venepuncture after an overnight fast (10 to 14 h). Serum was obtained by centrifugation at 4 °C for 20 min at 1800× *g*, immediately shipped to the accredited analysing lab or aliquoted and stored at −80 °C until further analyses. Triglycerides, HDL-cholesterol and glucose were measured by internationally validated standard enzymatic assays using automated modular blood analyser systems (ROCHE cobas 8000). For a full SOP, please refer to the open access protocol [18]. Aliquots of serum samples were used for fatty acid composition analysis (all volunteers) and quantitative analysis of polyphenol metabolites, short (SCFAs) and medium chain fatty acids (MCFAs), as well as untargeted metabolomics (dairy arm only).

### 2.9. Dietary Assessment

At baseline, participants were asked to complete a food frequency questionnaire (FFQ) designed to assess bioactive intake [18]. Bioactive intake was calculated using probabilistic exposure from frequency data and bioactive composition of foods [17].

### 2.10. Compliance

During the 28-day intervention period, participants were to keep a record of the BEF they consumed in a diary, particularly noting days where BEF were not consumed or only part consumed. Participants were also asked to return unused foods and empty packaging to the RC at the end of the trial period.

### 2.11. Clinical Data Collection and Management

All measures collected during the study were recorded in a pseudonymised form (according to randomization code) using the electronic data capture (EDC) system Mythos CDMS v2.0. The EDC system is hosted by AdWare Research Ltd. (Völgy u, Hungary). Personal identifying information was stored separately from the outcome measures.

### 2.12. Serum Fatty Acid Composition

Total lipids were extracted from serum according to Folch et al. with slight modifications [22]. Briefly, 2 mL of methanol, and 2 mL of chloroform were sequentially added to 0.2 mL of plasma followed by through mix with stirring magnet at the maximum intensity and incubation in a water bath for 20 min at 60 °C. Successively, 2 mL of chloroform were added and the solution filtered through a filter paper into a new test tube. Four mL of 0.1 M KCl was added to the sample and kept overnight at 4 °C. Then the lower chloroform layer was transferred to a test tube, and trans-methylation was performed [23]. Fatty acids composition (as methyl esters) was determined by GC (Trace GC Thermo with FID detector) using a capillary column (SP2340, 0.2 μm film thickness) with a programmed temperature gradient (60–240 °C, 4 °C/min), as previously reported [24]. The gas chromatographic peaks were identified on the basis of their retention time ratios relative to methyl stearate and predetermined by use of authentic samples. Gas chromatographic traces and quantitative evaluations were obtained using a Chrom-Card software computing integrator (Thermo Fisher Scientific Inc., Waltham, MA, USA).

### 2.13. Quantification of Short and Medium Chain Fatty Acids, and Polyphenol Metabolites in Serum

For analysis of SCFA/MCFA, a liquid–liquid extraction procedure was used [25]. For quantification a Trace GC Ultra gas chromatograph (Thermo Fisher Scientific, San Jose, CA, USA) was used, equipped with an autosampler PAL combi-xt autosampler (CTC Diagnostics AG, Zwingen, Switzerland) coupled to a TSQ Quantum XLS tandem mass spectrometer (Thermo Fisher Scientific, San Jose, CA, USA). The GC–MS data processing was performed using a qualitative and quantitative software package, XCALIBUR™ 2.2 (Thermo Fisher Scientific, San Jose, CA, USA).

For analysis of polyphenol metabolites, the extraction procedure was as described previously [26]. The MS system used was a Waters Xevo TQ (Milford, MA, USA) triple quadrupole mass spectrometer, coupled with an electrospray interface and polarity switching option during acquisition. The LC-MS data processing was performed using qualitative and quantitative packages in Waters MassLynx 4.1 and TargetLynx software.

### 2.14. Untargeted Metabolomics Analysis of Serum

Additionally, untargeted analysis of serum samples was described previously [27] using LC-HR Orbitrap LTQ-XL. Raw files were processed with Compound Discoverer 2.1 Software (Thermo Fisher Scientific, San Jose, CA, USA). Data processing included the following steps: alignment of chromatograms, pick peaking, m/z features grouping in compounds, subtraction of blanks and solvents, gap filling and normalization to TIC (total ion chromatogram—normalization to instrumental factor).

### 2.15. Statistical Analysis

The full analysis dataset (FAS) was defined as all randomized subjects who received experimental products and have primary outcome data both at baseline and at the end of the study. Descriptive statistical methods were applied including the case number and frequency for categorical variables, the case number, mean, standard deviation, standard error of mean, minimum, median and maximum values for continuous variables. Mean change in fasting blood TG and HDL-cholesterol (primary endpoints), secondary outcomes (fasting blood glucose, systolic and diastolic BP, WC) and serum fatty acid composition was analyzed using one-factor ANOVA followed by paired *t*-tests or Tukey tests depending on distribution. Statistical evaluation was done using IBM SPSS Statistics 19.0 (Armonk, NY, USA). The paired *t*-test with Bonferroni correction for false discovery rate was applied to data from SCFA, MCFA, polyphenol catabolites as well as to untargeted metabolite analysis. 

## 3. Results

### 3.1. Baseline Characteristics

Out of the 333 volunteers who were screened, 172 of them were eligible, 167 concluded the study and were included into the FAS dataset. Recruitment of participants into the egg (pancake) trial was more challenging compared to dairy and bakery, resulting in lower number of participants. This was attributed to lower familiarity with pancakes and the requirement for domestic frozen storage by participants. However, randomization per treatment was successful, since allocation to treatment groups within each matrix was comparable. The baseline demographic, anthropometric, metabolic characteristics of all participants who completed the trial is presented in Table 3. Differences at baseline in anthropometric or clinical were not statistically significant different between recruiting centers, between genders or between treatment allocations (95% confidence).

Bioactive exposure derived from FFQ data revealed that intake of all three bioactives by participants in all RC was very low (Table 4), and much lower (around 10 fold) than the levels of enrichment of BEF (Table 1). There was a similar intake of DHA in all three countries, while BG and AC intakes varied more widely. The higher intake of OBG in the egg arm is attributed to higher frequency of consumption of oat porridge and biscuits, commonly available in the UK, while the higher AC exposure was attributed to fruit squash and fruit yogurts. The exposure is still much lower that the dose provided by administration of BEF.

### 3.2. Acceptance of BEF and Compliance

BEF acceptability was assessed at T28 using a 9-point hedonic scale for appearance, odour, flavour, texture and overall acceptability (Table 5). The results suggest that the milkshakes were the most accepted products, followed by the pancakes and biscuits. Commonly, consumers found the purple colour of anthocyanins unfamiliar, but not-off putting. Panelists found the biscuits to be somewhat hard, resulting in lower texture score. BEF were generally well accepted during the trial with high levels of compliance (>85%) recorded in all RC (Table 5). There was no significant difference in compliance amongst the treatments or RC.

### 3.3. Primary Outcomes

Fasting TG and HDL-cholesterol were defined as primary endpoints and measured at baseline (T0) and at the end of the intervention period (T28) by standardized enzymatic methods. Mean change in blood TG and HDL-cholesterol is shown in Table 6. 

The results indicate that the dairy intervention had an effect on both primary endpoints, with a significant decrease in TG level in the DHA + OBG group and a significant increase in HDL-C in the AC group. The results of the egg intervention showed the largest overall raising effect on HDL-C, with a significant increase in the DHA and DHA + AC groups. The results of the bakery intervention showed the most consistent change in TG and HDL-C, with most arms lowering TG (apart from AC) and raising HDL-C, however none of the changes were statistically significant at 95% confidence level (paired *t*-tests).

### 3.4. Secondary Outcomes

Mean changes in each secondary end-points between baseline (T0) and endpoint (T28) are shown in Table 7. None of the treatments showed significant effects on the secondary outcomes at 95% confidence level (paired *t*-tests).

### 3.5. Serum Fatty Acid Composition and Quantification of SCFAs, MCFAs and Polyphenol Metabolites

Serum fatty acid composition was determined to verify bioavailability of DHA and volunteer compliance to treatment. At T0, the general pattern of fatty acid percent composition was similar amongst treatment groups within each RC (Table 8). Comparing serum fatty acid composition at T0 and T28 there was a significant percent increase in DHA (22:6n-3) with consumption of milkshake enriched with DHA and DHA + OBG, and of biscuits enriched in DHA + AC and DHA + OBG. Pancake consumption did not alter fatty acid composition significantly, though results showed similar size effects to the other matrices in treatments containing DHA, alone or in combination with other bioactives.

Statistical analysis did not reveal any significant difference in SCFA, MCFA, and polyphenol metabolite concentrations in serum of the participants in the dairy arm (Appendix A).

### 3.6. Adverse and Serious Adverse Effects

There were no serious adverse effects reported during the trial. Minor adverse effects were reported which were mainly unrelated to the trial (e.g., non-food associated illness). A ‘gastrointestinal symptoms questionnaire’ assessed self-reported changes to gastrointestinal function including loose stools/diarrhea, constipation, bloating, increase or decrease in appetite. Overall, none of the treatments has significant effects on gastrointestinal symptoms in any of the RC (data not shown).

## 4. Discussion

The results of the PATHWAY-27 pilot study demonstrated the feasibility of undertaking a dietary intervention with BEF. Interventions with foods are more difficult than interventions with supplements for both volunteers and researchers [13]. RCs face a higher burden in terms of storage and distribution of large parcels of perishable food, blinding of foods, collection of empty packaging for compliance. These were addressed in this pilot study through the design of a detailed food production plan by the trial coordinator, which detailed exact dates of food production and delivery to RCs and participants. Blinding was achieved through packaging of foods in opaque foil and labelling of foods with a specific code for each participant. 

Trials with food also provide challenges for participants who may struggle to eat the same food every day. We observed high levels of acceptance, completion rates and compliance in all RC which could be explained by the general sensory acceptability of BEF, as well the short duration of the trial (4 weeks). An important consideration is the cost of producing foods compared to supplements. While we did not undertake a detailed cost analysis, keeping costs low was considered in the design of the trial by selecting staple foods that are cheap to produce and have shelf-lives long enough to cover the trial period. Bioactive enriched foods may be more expensive than supplements [28]. However, a food-based approach is generally preferred because foods may contain more nutrients compared to supplements. BEF can bring cost-effective benefits to consumers.

The results from the bioactive exposure analysis concluded that habitual bioactive consumption amongst these European consumers is low. This has been observed for other bioactives such as catechins [5]. Low intakes from habitual diet justify an intervention to increase intake in this population. The enrichment of food with bioactives could lead to a ten-fold increase in the consumption of bioactives. As the trial did not have a healthy group, we cannot generalise low bioactive intake to the general population. However, the high compliance in this study suggests BEF consumption is a viable strategy to increase bioactive intake in MetS populations.

Results of the pilot study indicated that consumption of specific BEF has the potential to improve lipid metabolism, including markers of MetS. BEF effectiveness appears not solely dependent on the bioactive(s) used for enrichment, but also on the matrix. The matrix could have an impact on bioactive bioavailability, for example dairy has been suggested as a good vehicle for DHA delivery. Consumption of 500 mL/d n-3 PUFA-enriched milk has been reported to increase serum DHA concentration after four weeks, producing a significant decrease in TG level after eight weeks [28]. In agreement, we observed a significant increase in serum DHA content after 28 days consumption of dairy BEF enriched with DHA and DHA + OBG. In the DHA + OBG group, a significant decrease in TG level was also observed. These results concur with previous evidence that dairy is a suitable matrix for PUFA delivery [29]; of note, no modifications in serum DHA concentration were detected in volunteers consuming DHA + AC enriched milkshake. Since compliance was similar to other enrichments, this suggests that AC may impact on availability of DHA. It has already been reported that phenolic compounds could interact negatively with fat-soluble vitamin absorption [30,31] but there are no data in the literature on interaction with fatty acid absorption. Of note, the putative antagonistic effect of AC on DHA bioavailability was not observed when the bioactives were delivered within the other two matrices. 

In our study, it was not possible to verify differences in OBG and AC bioavailability after milkshake consumption OBG and AC metabolites, SCFA and polyphenol secondary metabolites, respectively, were found at low concentrations (Appendix A and Appendix A respectively). Blood samples were collected in fasting condition, i.e., many hours after BEF consumption, this may explain the low levels of SCFA (putative marker for OBG consumption) and polyphenol metabolites (putative maker for AC consumption) concentration after the dairy intervention. In rats, clearance of polyphenol metabolites and SCFA is very rapid, and plasma level returned close to baseline within a few hours of consumption [32]. We did not directly evaluate AC and OBG bioavailability and metabolism in the bakery and egg groups. Notwithstanding, the effect of the food matrix on bioaccessibility of bioactives in the different BEF had been previously investigated in vitro, indicating that accessibility of AC from biscuits was limited [33], and solubilisation of AC from milkshake was quicker than from pancakes during simulated in vitro digestion [34].

Besides bioavailability, bioactives could be acting synergistically or antagonistically on metabolism, and synergism and antagonism are largely unknown and unexplored. It is known that different bioactives have similar metabolic endpoint effects, acting on different metabolic pathways in different cells [16,24,35,36]. Regarding lipid metabolism, DHA may act on hepatic and adipose cells via modulation of the transcriptome and proteome [24,36] while the effect of AC could be via regulation of lipoprotein lipase activity [37] or cholesterol ester transfer protein [7]. The lipid lowering effect of OBG is usually related to increased rate of bile acid excretion [38,39]. However, OBG metabolism by the gut microflora to SCFAs and MCFAs may also contribute to its positive effects on health [40]. In turn, the intestinal microbiota also biotransform the majority of dietary AC into active and bioavailable metabolites [41]. Differences in gut microbiota ecology determine polyphenol metabolism, producing different metabotypes that could modulate health effects [42]. In this study, we did not consider the impact of genotype, microbiotype, physical activity or dietary intake (apart from bioactive intake) as confounding factors. The lack of a placebo group was also a limitation. However, the trial was successful as proof-of-concept that BEF can modulate metabolism. Based on this pilot study, a larger, placebo controlled randomized controlled trial will further investigate the impact of BEF consumption on MetS markers. The larger study will also consider the impact of genotype, microbiotype, physical activity and dietary intake on observed metabolic changes.

In conclusion, the results of the present pilot study give preliminary insights into the complex and often neglected interactions between the food matrix and bioactives used for enrichment allowing a more holistic selection of the best enrichment for each matrix to be tested in a larger, longer and placebo controlled intervention trial. The lessons learned from this pilot studies have been compiled as a practical guide to help researchers design and implement pilot studies for nutritional science [43].

## Figures and Tables

**Figure 1 nutrients-11-01814-f001:**
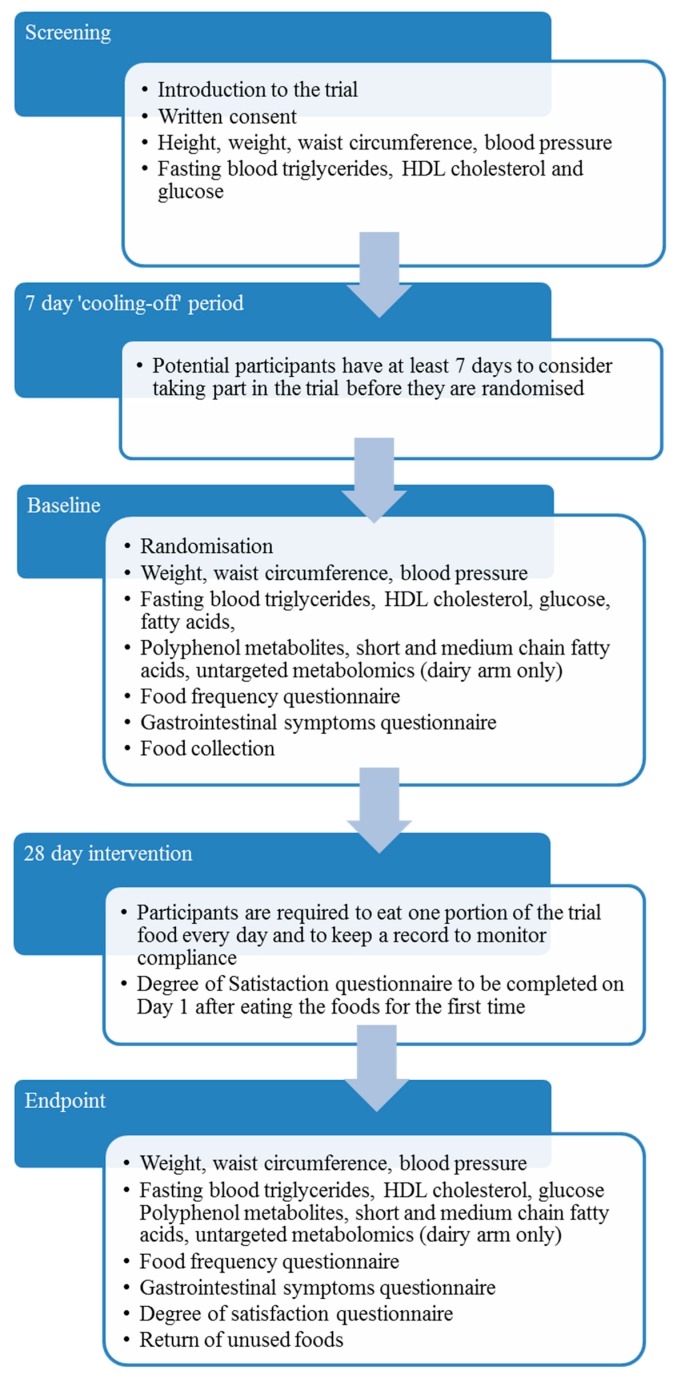
Trial design and summary of participant activities during the PATHWAY-27 pilot studies. HDL = high density lipoprotein.

**Table 1 nutrients-11-01814-t001:** Dose of bioactives in each treatment arm tested in the PATHWAY-27 pilot study.

Bioactive	Treatment Arm				
	DHA	OBG	AC	DHA + OBG	DHA + AC
DHA (mg)	250	-	-	250	250
OBG (mg)	-	3000	-	3000	-
AC (mg)	-	-	320	-	320

DHA = docosahexaenoic acid, OBG = oat beta-glucan and AC = grape anthocyanins.

**Table 2 nutrients-11-01814-t002:** Energy and macronutrient content (median of all BEF, daily serving).

	Bakery	Dairy	Egg
Energy (kJ)	828 (657–966)	904 (653–929)	854 (711–983)
Energy (kcal)	198 (157–231)	216 (156–222)	204 (170–235)
Protein (g)	6.1 (4.1–6.2)	13.9 (9.6–16.3)	5.6 (4.3–6.5)
Carbohydrate (g)	35.2 (22.2–36.9)	26.7 (25.2–28.9)	17.7 (14.1–22.0)
of which sugars (g)	9.3 (6.7–10.0)	22.1 (20.0–22.7)	1.9 (1.7–11.4)
Fat (g)	7.6 (4.6–7.7)	4.4 (1.8–4.6)	13.1 (9.3–14.3)
of which SFA (g)	1.6 (0.9–1.7)	1.5 (1.0–1.6)	2.2 (1.3–2.4)
Fibre (g)	0.8 (0.7–5.2)	0.8 (0.7–5.2)	0.7 (0.7–4.5)
Sodium (mg)	10.9 (4.0–13.5)	135.3 (123.7–139.6)	413.0 (397.7–617.7)

BEF = Bioactive enriched food, SFA = saturated fatty acids.

**Table 3 nutrients-11-01814-t003:** Participant demographic, anthropometric, metabolic and bioactive exposure data at baseline of the PATHWAY-27 pilot study. Mean (SD).

	Matrix					
	Dairy		Egg		Bakery	
Gender	M	F	M	F	M	F
N	38	28	14	21	22	44
Age (years)	61.6 (9.0)	58.9 (7.8)	45.5 (10.7)	44.6 (14.2)	51.8 (13.1)	56.4 (12.6)
Weight (kg)	91.2 (13.6)	80.9 (12.9)	99.3 (18.3)	87.3 (22.0)	91.4 (16.0)	80.8 (12.4)
BMI (kg/m^2^)	29.26	29.97	31.11	31.68	29.31	30.80
Waist circumference (cm)	105.3 (11.2)	98.2 (10.5)	110.1 (9.4)	104.8 (16.0)	101.6 (12.1)	93.2 (8.3)
Fasting TG (mg/dL)	187.0 (94.9)	162.4 (53.3)	155.6 (54.0)	136.6 (84.5)	167.7 (50.89)	195.41 (122.26)
Fasting HDL-C (mg/dL)	44.0 (10.0)	53.5 (10.9)	40.3 (6.7)	55.2 (16.6)	40.0 (11.63)	48.1 (13.3)
Fasting Glucose (mg/dL)	96.8 (10.2)	91.6 (8.1)	93.3 (14.4)	91.1 (9.7)	104.4 (16.3)	97.6 (9.9)
Systolic BP (mmHg)	144.1 (19.8)	140.3 (16.2)	134.7 (21.3)	130.7 (17.0)	139.4 (17.0)	133.7 (11.2)
Diastolic BP (mmHg)	91.9 (11.5)	93.6 (10.8)	79.5 (14.1)	82.1 (11.4)	86.9 (8.2)	85.3 (9.8)

SD = standard deviation, M = male, F = female, BMI = body mass index, TG = triglycerides, HDL-C = high density lipoprotein cholesterol, BP = blood pressure.

**Table 4 nutrients-11-01814-t004:** Bioactive exposure at baseline of participants in each recruiting centre showing mean (SD).

Bioactive	Germany (Dairy Arm)	UK (Egg Arm)	France (Bakery Arm)
DHA (mg/d)	0.31 (0.24)	0.34 (0.28)	0.36 (0.50)
OBG (mg/d)	290.5 (473.78)	716.5 (1153.41)	154.1 (387.83)
AC (mg/d)	46.3 (39.39)	102.2 (92.67)	49.2 (74.49)

DHA = docosahexaenoic acid, OBG = oat beta-glucan and AC = grape anthocyanins.

**Table 5 nutrients-11-01814-t005:** Sensory acceptance scores of BEF (measured using 9-point hedonic scales) and compliance to the trial according to matrix.

Attribute	Matrix		
	Dairy	Egg	Bakery
Appearance	6.80	5.84	5.37
Odour	6.86	5.72	5.79
Taste	6.98	5.65	5.18
Texture	6.80	5.42	4.22
Overall acceptability	7.03	5.88	4.75
Compliance to trial (%)	93.3	90.6	85.2

**Table 6 nutrients-11-01814-t006:** Mean change and standard deviation (SD) in fasting serum triglycerides (TG) and HDL-C primary outcomes from baseline (T0) to endpoint (T28) according to matrix and bioactive enrichment.

			TG (mg/dL)	HDL-C (mg/dL)
Matrix	Enrichment	N	Mean (SD)	Mean (SD)
Dairy	DHA	14	10.36 (60.33)	−2.36 (7.96)
	OBG	12	−31.00 (128.64)	−0.17 (8.36)
	AC	15	−7.60 (50.39)	2.53 * (3.89)
	DHA + OBG	13	−41.69 * (50.77)	1.08 (4.21)
	DHA + AC	12	−4.83 (49.76)	0.17 (6.09)
Egg	DHA	7	−6.34 (45.96)	4.97 * (5.34)
	OBG	8	21.04 (89.28)	0.46 (5.62)
	AC	8	−13.16 (28.65)	−0.46 (6.02)
	DHA + OBG	6	0.00 (36.72)	−1.92 (5.33)
	DHA + AC	6	−11.83 (37.84)	5.80 * (4.04)
Bakery	DHA	12	−7.75 (51.34)	2.83 (2.66)
	OBG	14	−6.86 (49.96)	1.00 (6.15)
	AC	13	32.00 (115.00)	4.82 (1.34)
	DHA + OBG	13	−27.54 (46.13)	1.15 (5.65)
	DHA + AC	14	−36.64 (82.32)	1.43 (4.52)

* Shows statistical difference at 95% confidence between baseline and endpoint within the enrichment arm using post-hoc paired *t*-tests for each participant.

**Table 7 nutrients-11-01814-t007:** Mean change and standard deviation (SD) in systolic BP, diastolic BP, fasting blood glucose and waist circumference (WC) secondary outcomes from baseline (T0) to endpoint (T28) according to matrix and bioactive enrichment.

			Systolic BP (mmHg)	Diastolic BP (mmHg)	Fasting Glucose (units)	WC (cm)
Matrix	Enrichment	N	Mean (SD)	Mean (SD)	Mean (SD)	Mean (SD)
Dairy	DHA	14	0 (11)	−2 (7)	−0.9 (7.8)	−0.6 (1.7)
	OBG	12	−9 (11)	−7 (12)	1.3 (4.9)	0.4 (1.6)
	AC	15	2 (11)	2 (7)	−2.2 (5.1)	−0.6 (2.2)
	DHA + OBG	13	−3 (12)	−4 (8)	0.8 (3.0)	−1.3 (2.3)
	DHA + AC	12	−3 (16)	−1 (10)	0.5 (6.9)	1.1 (2.0)
Egg	DHA	7	3 (10)	1 (7)	4.5 (6.5)	−4.2 (9.0)
	OBG	8	0 (9)	2 (10)	0.0 (7.5)	−0.8 (7.1)
	AC	8	6 (9)	6 (2)	−4.2 (7.9)	−2.7 (4.6)
	DHA + OBG	6	−3 (11)	0 (13)	−2.4 (4.1)	−1.8 (2.5)
	DHA + AC	6	−12 (10)	5 (8)	−2.2 (15.1)	−3.6 (5.5)
Bakery	DHA	12	1 (9)	−5 (5)	−1.17 (5.18)	−0.83 (2.41)
	OBG	14	−2 (10)	−2 (8)	−1.21 (3.31)	0.43 (1.60)
	AC	13	3 (14)	2 (6)	0.15 (6.62)	−0.36 (2.10)
	DHA + OBG	13	0 (10)	0.3 (7)	−2.62 (7.40)	−0.62 (1.98)
	DHA + AC	14	3 (11)	−4 (10)	−1.15 (4.86)	0.15 (2.08)

**Table 8 nutrients-11-01814-t008:** Fatty acid composition (mol/100 mol) in different groups at T0 and T28.

	**Dairy BEF (Milkshake)**									
	**DHA**		**AC**		**OBG**		**DHA + AC**		**DHA + OBG**	
	**T0**	**T28**	**T0**	**T28**	**T0**	**T28**	**T0**	**T28**	**T0**	**T28**
14:0	0.9 ± 0.5	0.9 ± 0.5	1.1 ± 0.4	1.2 ± 0.6	1.2 ± 0.4	1.1 ± 0.4	0.9 ± 0.4	0.9 ± 0.4	1.4 ± 0.6	1.1 ± 0.6
16:0	27.9 ± 1.8	30.5 ± 8.9	28.8 ± 3.4	29.2 ± 3.2	28.2 ± 2.3	31.7 ± 8.5	28.9 ± 4	28 ± 3.2	27.7 ± 2.3	28.3 ± 2.4
16:1	1.5 ± 0.6	2 ± 0.9	1.8 ± 0.5	1.9 ± 0.8	2.1 ± 1	2.2 ± 1.4	1.3 ± 0.5	1.4 ± 0.6	2.4 ± 1.1	2.3 ± 2.4
18:0	10.4 ± 2.2	10.4 ± 3.4	9.3 ± 2.1	9.0 ± 1.3	9.5 ± 2.4	10.2 ± 2.1	10.1 ± 2.1	10.5 ± 2.8	9.1 ± 2.1	9.9 ± 1.5
18:1	22.7 ± 2.2	25.1 ± 7.3	23.5 ± 2.8	23.9 ± 1.8	25.5 ± 2.8	28.1 ± 7	22.3 ± 3.1	22.6 ± 3.5	24.9 ± 3.8	22.4 ± 4
18:2	24.9 ± 3.3	26.3 ± 9.5	23.6 ± 3.8	23.9 ± 3.7	23.5 ± 4	25.3 ± 6.7	24.3 ± 4.9	24.3 ± 3	23.7 ± 4.8	23.8 ± 5.2
18:3	0.3 ± 0.3	0.2 ± 0.2	0.2 ± 0.3	0.2 ± 0.2	0.1 ± 0.1	0.3 ± 0.3	0.2 ± 0.2	0.4 ± 0.7	0.2 ± 0.2	0.1 ± 0.2
20:4	7.3 ± 1.6	6.3 ± 2	7.1 ± 1.6	6.6 ± 1.4	5.5 ± 1.4	7 ± 2.5	7.1 ± 1.8	6.9 ± 1.5	6.5 ± 1.8	7 ± 1.3
20:5n-3	0.5 ± 0.4	0.5 ± 0.6	0.4 ± 0.4	0.5 ± 0.5	0.3 ± 0.4	0.3 ± 0.4	0.6 ± 0.4	1.1 ± 2	0.5 ± 0.5	0.6 ± 0.7
22:6n-3	3.6 ± 0.8	5.3 ± 2.8 *	4.1 ± 1.2	3.7 ± 1.1	4.1 ± 0.9	3.7 ± 1	4.2 ± 1.1	3.9 ± 0.9	3.6 ± 1.0	4.4 ± 1.4 *
	**Egg-based BEF (Pancake)**									
	**DHA**		**AC**		**OBG**		**DHA + AC**		**DHA + OBG**	
	**T0**	**T28**	**T0**	**T28**	**T0**	**T28**	**T0**	**T28**	**T0**	**T28**
14:0	0.4 ± 0.4	0.30 ± 0.3	0.736 ± 0.63	0.4 ± 0.3	0.8 ± 0.1	0.9 ± 0.1	0.4 ± 0.1	0.4 ± 0.2	0.3 ± 0.2	0.4 ± 0.2
16:0	25.5 ± 4.0	27.0 ± 3.5	27.1 ± 4.1	24.6 ± 2.9	27.2 ± 2.6	27.2 ± 2.6	26.1 ± 0.3	26.1 ± 0.4	26.9 ± 1.7	27.3 ± 1.8
16:1	1.4 ± 0.7	1.2 ± 0.8	1.7 ± 0.0	1.3 ± 0.7	0.9 ± 0.8	0.6 ± 0.2	1.1 ± 0.5	0.8 ± 0.1	1.5 ± 0.3	1.3 ± 1.3
18:0	9.1 ± 1.3	10.4 ± 2.4	8.4 ± 1.8	10.0 ± 1.3	10.0 ± 0.1	10.2 ± 0.2	8.6 ± 1.8	8.9 ± 0.9	9.2 ± 1.1	11.3 ± 2.9
18:1	24.7 ± 0.7	22.3 ± 2.3	24.0 ± 1.4	23.7 ± 1.5	22.9 ± 2.5	20.4 ± 0.4	24.8 ± 1.9	22.4 ± 0.9	24.8 ± 5.5	19.3 ± 2.0
18:2	28.9 ± 4.3	28.7 ± 7.2	28.0 ± 0.2	28.9 ± 1.7	27.3 ± 3.3	28.9 ± 3.7	28.6 ± 0.9	30.8 ± 1.1	26.7 ± 2.4	25.3 ± 3.8
18:3	0.1 ± 0.1	n.d.	n.d.	n.d.	n.d	n.d.	0.1 ± 0.2	n.d.	0.1 ± 0.1	0.1 ± 0.2
20:4	6.2 ± 0.7	5.1 ± 2.1	6.1 ± 3.8	7.3 ± 4.6	6.3 ± 0.7	7.3 ± 0.2	6.3 ± 0.1	5.9 ± 0.4	6.9 ± 1.2	8.8 ± 0.9
20:5n-3	0.3 ± 0.3	0.2 ± 0.2	0.2 ± 0.0	n.d.	0.6 ± 0.4	0.4 ± 0.2	0.1 ± 0.1	0.2 ± 0.3	0.7 ± 0.9	1.2 ± 0.4
22:6n-3	3.3 ± 0.7	4.9 ± 1.7	3.7 ± 1.1	3.9 ± 1.1	4.2 ± 0.1	4.0 ± 1.0	3.9 ± 0.1	4.4 ± 0.2	3.1 ± 0.9	4.9 ± 1.1
	**Bakery BEF (Biscuits)**									
	**DHA**		**AC**		**OBG**		**DHA + AC**		**DHA + OBG**	
	**T0**	**T28**	**T0**	**T28**	**T0**	**T28**	**T0**	**T28**	**T0**	**T28**
14:0	1.4 ± 0.3	1.5 ± 0.3	1.7 ± 0.4	1.7 ± 0.4	1.4 ± 0.6	1.3 ± 0.5	1.8 ± 0.7	1.6 ± 0.6	1.9 ± 0.6	1.6 ± 0.6
16:0	27.0 ± 0.8	25.9 ± 1.9	26.6 ± 2.0	26.5 ± 1.7	26.6 ± 2.0	25.9 ± 2.6	27.8 ± 1.8	27.3 ± 0.9	28.6 ± 3.1	26.7 ± 1.9 *
16:1	3.1 ± 0.8	2.9 ± 0.7	3.0 ± 0.7	3.0 ± 0.6	2.6 ± 0.8	2.6 ± 0.9	3.3 ± 0.70	3.3 ± 0.56	3.6 ± 0.5	3.1 ± 1.0
18:0	9.4 ± 1.7	8.9 ± 1.2	9.3 ± 1.5	8.9 ± 1.1	8.6 ± 1.6	8.4 ± 0.9	9.0 ± 1.3	8.6 ± 1.9	9.0 ± 1.1	8.7 ± 0.9
18:1	25.2 ± 0.9	23.1 ± 2.4	23.4± 2.5	24.8 ± 3.4	24.2 ± 1.8	25.0 ± 2.4	24.6 ± 4.3	25.1 ± 3.0	24.1 ± 0.8	24.4 ± 2.7
18:2	28.6 ± 3.0	26.2 ± 2.1	26.0 ± 4.0	25.3 ± 3.3	26.3 ± 3.5	26.0 ± 4.0	23.8 ± 4.3	22.9 ± 2.4	22.7 ± 3.4	23.6 ± 4.2
18:3	1.3 ± 0.4	1.2 ± 0.1	1.1 ± 0.3	1.2 ± 0.3	1.0 ± 0.3	1.1 ± 0.3	1.1 ± 0.2	0.9 ± 0.1	1.1 ± 0.2	1.0 ± 0.3
20:4	6.3 ± 0.9	6.9 ± 1.3	6.2 ± 1.0	6.1 ± 1.1	6.3 ± 1.0	6.9 ± 1.4	6.1 ± 1.7	6.7 ± 1.3	6.1 ± 0.8	6.8 ± 1.2
20:5n-3	0.9 ± 0.3	0.9 ± 0.3	0.9 ± 0.5	0.8 ± 0.3	0.8 ± 0.2	0.8 ± 0.2	0.7 ± 0.1	0.9 ± 0.3	0.8 ± 0.2	1.0 ± 0.3
22:6n-3	2.3 ± 0.5	2.5 ± 0.7	1.9 ± 0.3	1.8 ± 0.3	2.1 ± 0.5	2.0 ± 0.5	1.8 ± 0.3	2.6 ± 0.5 **	1.9 ± 0.7	3.1 ± 1.1 *

Data are means ± SD and expressed as mol/100 mol. In the same group differences between T0 and T28 were tested for statistical significance by paired Student’s *t* tests (* *p* < 0.05; ** *p* < 0.001). n.d. = not detected.

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
