# Peer review of "A Dietary Intervention of Bioactive Enriched Foods Aimed at Adults at Risk of Metabolic Syndrome: Protocol and Results from PATHWAY-27 Pilot Study"

_nutrients, 2019, doi:10.3390/nu11081814_

Round 1
Reviewer 1 Report
Overview
The authors examined the effects on consumer acceptability, gastrointestinal tolerance, blood pressure, waist circumference, and various serum chemistries of bioactive enriched foods (BEF) in 167 individuals with various aspects of metabolic syndrome (MetS). The authors report that the foods were well-accepted and tolerated and were associated with favorable but largely not statistically significant changes in serum triglycerides, HDL-c, systolic and diastolic BP, fasting glucose, and waist circumference.
Major Comments
This reviewer likes the approach of providing these bioactive compounds incorporated into food rather than as separate supplements in pills. Nevertheless, this food-based approach must be more expensive than the pill-based approach. Assuming that this assumption is true, can the authors provide justification for the greater cost of this approach, possibly with greater expected outcomes with lower treatment costs for MetS?
How was the dose of each BEF determined?
The authors report that although the outcomes trended in favorable directions, none of the outcome measurements were statistically significant. Given this, can the authors says that the interventions were "effective"?
The authors report that BEF intake was "low" among study participants. Is this BEF intake comparable to intake for the general population? In other words, is BEF uncharacteristically low in this study population, compared to what this reviewer suspects is also a low intake for the general population?
The preceding questions raises another question. Is the MetS of the participants due to their low BEF intake? Conversely, is BEF intake for the population at large low and these participants have other concomitant factors that combine with low BEF intake to yield MetS?
Minor Comments
6. What was the ethnic make up of the study participants?
Author Response
We thank the reviewer for taking the time to review our manuscript.
Major Comments
This reviewer likes the approach of providing these bioactive compounds incorporated into food rather than as separate supplements in pills.
We thank the reviewer for this comment.
Nevertheless, this food-based approach must be more expensive than the pill-based approach. Assuming that this assumption is true, can the authors provide justification for the greater cost of this approach, possibly with greater expected outcomes with lower treatment costs for MetS?
We thank the reviewer for the interesting comment. While have not undertaken a detailed cost analysis due to low production volume for trial compared to large commercial operations (which would over estimate food cost), we have made a number of decisions to keep costs low, including:
· We have purposely chosen foods that are cheap to produce (biscuits: flour, sugar, oil; pancakes: flour, milk, egg; milkshakes: low-fat milk mainly); they were manufactured at pilot scale in a factory already producing these types of foods at a very low cost. The cost of producing these bioactive-enriched foods is not higher than producing supplements, which have to be enrobed and packaged.
· We have chosen bioactive ingredients which are already commercially produced at scale.
· A considerable amount of effort was dedicated through the project to produce foods with long shelf lives. The biscuits and the milkshakes were stable for the duration of the trial at room temperature. Therefore no increased cost compared to supplement. The pancakes had to be frozen for food safety reasons and were stable for the duration of the trial. Therefore, the only additional cost was from cold transport and storage (around € 6 per month).
Furthermore, food-based and pill-based approaches should not be simply compared based on costs. In all countries, Dietary Guidelines state that nutritional needs should be met primarily through the diet. Supplements aren't intended to substitute food. Although they might be appropriate in some situations they can't replicate all of the nutrients and benefits of whole foods. Watters et al (CA Watters, CM Edmonds, L Rosner, KP Sloss, PS Leung. A Cost Analysis of EPA and DHA in Fish, Supplements, and Foods. J Nutr Food Sci 2012, 2:8) has produced detailed information on the relative cost of long-chain omega-3 fatty acids available from common fish, seafoods, supplements, and fortified foods, concluding that added expense of obtaining n-3 LC-PUFAs in fortified products may be justified for individuals unwilling or unlikely to consume seafood or supplements on a regular basis.
We have also added a sentence to the discussion to discuss cost implications (line 370-376).
How was the dose of each BEF determined?
We have added a section in the materials and methods section to justify the doses (line 160).
The authors report that although the outcomes trended in favorable directions, none of the outcome measurements were statistically significant. Given this, can the authors says that the interventions were "effective"?
There were statistical improvements in primary outcomes according to matrix/bioactive, see table 6.
The authors report that BEF intake was "low" among study participants. Is this BEF intake comparable to intake for the general population? In other words, is BEF uncharacteristically low in this study population, compared to what this reviewer suspects is also a low intake for the general population?
We calculated the intake of bioactive components (i.e. beta-glucan, anthocyanin) and we evidenced that it was low in the studied population. As we did not have a healthy control group, we cannot generalize the intake to the general population. We have added a sentence in the discussion to this effect (line 402).
The preceding questions raises another question. Is the MetS of the participants due to their low BEF intake? Conversely, is BEF intake for the population at large low and these participants have other concomitant factors that combine with low BEF intake to yield MetS?
We think these are very interesting question. As MetS is due to a wide range of factors that lead to metabolic dysregulation, it is unlikely that MetS is purely due to low bioactive intake. Conversely, increased intake of some bioactives could help prevent MetS.
Minor Comments
6. What was the ethnic make up of the study participants?
Sorry, we did not collect this information. In our further trial, we will be collecting ethnicity and genotype.
Reviewer 2 Report
The research is current and interesting to nutrition area, but some parts of manuscrip should to improve.
In the text, references does not comply with the journal requirements (please check https://www.mdpi.com/journal/nutrients/instructions)
Introduction section:
- Where is de aim of study? The aim should go in the final paragraph of introduction section.
- Lines 45-47 and 48. It´s necessary a reference.
- In introduction section, it´s not necessary tables. Authors should be a reference rather than table 1 and change the wording of lines 47-48.
o Reference: Alberti KGMM, Eckel RH, Grundy SM, Zimmet PZ, Cleeman JI, Donato KA, et al. Harmonizing the metabolic syndrome: a joint interim statement of the International Diabetes Federation Task Force on Epidemiology and Prevention; National Heart, Lung, and Blood Institute; American Heart Association; World Heart Federation; International Atherosclerosis Society; and International Association for the Study of Obesity. Circulation 2009;120:1640–5. doi:10.1161/CIRCULATIONAHA.109.192644.
- Lines 83-96 should go in the material and methods section.
Material and methods section:
- In 2.7 Anthropometric measurements change to 2.7 Anthropometric measurements and clinical data. Blood preasure it´s not an anthropometric measurement. In addition, authors should indicate the accuracy (precision) of anthropometric material and they calculated Body Mass Index (BMI).
- 2.9. Dietary Assessment: Did authors take into account the nutritional intake and physical activity for the intervention?
Discussion section:
- Did authors take into account the nutritional intake and physical activity for the intervention?
- It´s necessary a paragraph of study limitations before a conclusions section (final paragraph of discussion section).

Author Response
The research is current and interesting to nutrition area, but some parts of manuscrip should to improved.
We thank the reviewer and have made some improvements to the manuscript according the feedback.
In the text, references does not comply with the journal requirements (please check https://www.mdpi.com/journal/nutrients/instructions)
We have corrected the formatting of the references.
Introduction section:
- Where is the aim of study? The aim should go in the final paragraph of introduction section.
We have modified the final paragraph of the introduction to make the aim more obvious (line 86-92).
- Lines 45-47 and 48. It´s necessary a reference.
We have removed the first sentence, we agree it makes it clearer.
- In introduction section, it´s not necessary tables. Authors should be a reference rather than table 1 and change the wording of lines 47-48.
We have already referenced Alberti et al, and we agree that there is no need to replicate information. We have removed table 1.
o Reference: Alberti KGMM, Eckel RH, Grundy SM, Zimmet PZ, Cleeman JI, Donato KA, et al. Harmonizing the metabolic syndrome: a joint interim statement of the International Diabetes Federation Task Force on Epidemiology and Prevention; National Heart, Lung, and Blood Institute; American Heart Association; World Heart Federation; International Atherosclerosis Society; and International Association for the Study of Obesity. Circulation 2009;120:1640–5. doi:10.1161/CIRCULATIONAHA.109.192644.
- Lines 83-96 should go in the material and methods section.
We have improved the last paragraph of the introduction to make the aim clearer. We have kept the description of the Pathway-27 project as readers will be able to look for further information about the project beyond this pilot study.
Material and methods section:
- In 2.7 Anthropometric measurements change to 2.7 Anthropometric measurements and clinical data. Blood preasure it´s not an anthropometric measurement. In addition, authors should indicate the accuracy (precision) of anthropometric material and they calculated Body Mass Index (BMI).
We have changed the heading to ‘Blood pressure and anthropometric measurement’. We have referred to the WHO guidelines and SOP in the open access protocol. We have indicated precision of measurement.
- 2.9. Dietary Assessment: Did authors take into account the nutritional intake and physical activity for the intervention?
In this pilot study, a specific FFQ was designed and used to monitor the bioactive intake. Energy and nutrient intake and physical activity will be evaluated in the larger intervention trial following the pilot study (lines 392-398).
Discussion section:
- Did authors take into account the nutritional intake and physical activity for the intervention?
As explained above, we have not evaluated the quality of the diet and the level of physical activity in volunteers of this pilot study, which aimed to select the BEF to be used in a larger intervention trial. These parameters will be assessed in the larger intervention trial (lines 392-398).
- It´s necessary a paragraph of study limitations before a conclusions section (final paragraph of discussion section).
We have added a paragraph outlining the limitations and the improvements that will be made in the design of a larger trial (lines 392-398)..
Round 2
Reviewer 1 Report
The authors have satisfactorily address my critiques.
Reviewer 2 Report
No comments to add, authors have included the proposed changes.